# Peer review of "The Application of Lophatherum Gracile Brongn Flavonoids in Wheat Flour Products: Effects on the Structural and Functional Characteristics of Wheat Dough"

_foods, 2024, doi:10.3390/foods13162556_

Round 1

Reviewer 1 Report

Comments and Suggestions for Authors

Topic 3.1 only presents a description of the results found and discussion, but does not cite other articles in the literature. As the authors previously reported that there are few studies on lophath, flavonoid erum gracile brongn, it is understood that there should be no reports in the literature. If this is the reason, I suggest you state this fact at the end of the paragraph. If you have comparative data, I suggest including it.

In lines 190 and 191, it mentions table 2, but it is actually table 1.

Topic 3.4 also does not present a discussion with other articles.

The authors describe in the conclusion the main effects of adding Lophatherum gracile brongn flavonoids to the dough. However, there was no conclusion about the feasibility of this addition, whether the effect would be beneficial or harmful industrially.

Author Response

Comments 1: Topic 3.1 only presents a description of the results found and discussion, but does not cite other articles in the literature. As the authors previously reported that there are few studies on lophath, flavonoid erum gracile brongn, it is understood that there should be no reports in the literature. If this is the reason, I suggest you state this fact at the end of the paragraph. If you have comparative data, I suggest including it.

Response 1: Thank you for your valuable feedback on our manuscript. We appreciate your thorough review and suggestions. Regarding Topic 3.1, we acknowledge your concern that it primarily presents the results and discussion without referencing other literature on lophatherum gracile brongn flavonoids. As we have previously noted, there are indeed limited studies available on this specific topic in the literature.

To address this, we clarified in the paragraph that the scarcity of existing literature on lophatherum gracile brongn flavonoids was the reason comparative data from other studies are not included. Additionally, we considered the suggestion to state this fact explicitly at the end of the paragraph to provide transparency to the reader:

Currently, there have been no studies reported on the influence of lophatherum gracile brongn flavonoid on the microstructure of dough. --Line 171

Comments 2: In lines 190 and 191, it mentions table 2, but it is actually table 1.

Response 2: Thank you for your keen observation. We appreciate you pointing out the discrepancy regarding the mention of Table 2 on lines 190 and 191, where it should correctly refer to Table 1. We revised this error in the manuscript accordingly.

Comments 3: Topic 3.4 also does not present a discussion with other articles.

Response 3: Thank you for your constructive feedback on our manuscript. Regarding Topic 3.4, we appreciate your observation that the discussion lacks engagement with other relevant articles. In our revised version, we expanded the discussion to include comparisons with existing literature. This provided a clearer context for our findings and highlight their contribution to the field:

Zhang studied the impact of tannic acid addition on dough structure, revealing that tannic acid, due to its reducing properties, disrupted disulfide bonds in wheat dough[32]. --Line 282

Comments 4: The authors describe in the conclusion the main effects of adding Lophatherum gracile brongn flavonoids to the dough. However, there was no conclusion about the feasibility of this addition, whether the effect would be beneficial or harmful industrially.

Response 4: Thank you for your insightful comments regarding the conclusion of our manuscript. We would like to clarify that in the introduction, we have extensively discussed various studies demonstrating the health benefits of adding Lophatherum gracile brongn flavonoids. Therefore, our conclusion emphasized the potential beneficial effects of this addition for industrial applications:

Phenolic compounds act as antioxidants by scavenging free radicals, thus poten-tially preventing chronic diseases. Studies have shown that incorporating phenol-rich substances in dough-based products significantly enhances their antioxidant capacity [15]. Methods to improve the bioavailability and utilization of phenolic compounds in dough products include incorporating phenol-rich fractions like bran, adding promis-ing natural phenolic substances or extracts, and using enzymes or mechanical strength to release bound phenolic compounds. Lophatherum gracile Brongn, a phenolic com-pound derived from natural plant extracts, possesses various physiological activities such as antioxidant, antimicrobial, and anticancer properties, and has been widely uti-lized in functional food applications [4-7]. However, research on the impact of lophatherum gracile Brongn on wheat dough remains scarce. This study aimed to in-vestigate the feasibility of incorporating bamboo leaf flavonoids into wheat dough and explore their effects on dough structure and functional properties. --Line 317-329

Reviewer 2 Report

Comments and Suggestions for Authors

In this work the authors discuss the application of lophatherum gracile brongn flavonoid in wheat flour product and investigate the effect on the structural and functional characteristics of wheat dough. The topic is interesting and the manuscript has a potential from a scientific point of view. However, there are some flaws that require the authors’ attention.

Introduction:

1.      In general it is well written. However, more information should be added about wheat dough, its importance and its properties (mechanical, nutritional, technological, etc). After all these matters are mentioned in the title.

2.      The aim paragraph is too long. It should be edited so the specifics of this research are more clear and precise. There is no need to be too technical about the analytical tools used in this study, in the aim paragraph.

Materials and methods:

1.      Line 74: Not familiar with that supermarket brandname. Give the website link (if available) or even better the coordinates of this store.

2.      Line 76: Add the city and country of origin of that company. The same applies for all equipment and materials used in this study.

3.      Despite detailed, subsections 2.3, 2.4 and 2.7 could use appropriate references. If however, the methods described here are newly developed, then calibration or reference curves and other crucial information for the method evaluation should be included in the discussion.

Results & Discussion:

1.      In general, specific parts of the discussion are well written and informative. However, subsections 3.1, 3.3 & 3.4, despite having some interesting results, are mostly oriented to describe results and no real discussion is provided. This is also depicted by the very few papers shown in the reference section of the manuscript. The discussion should be expanded so that the authors’ findings (and their importance) could be linked to existing literature. The authors are encouraged to compare their results to the action of other additives (ideally flavonoids) used in wheat dough. Subsections 3.2 and 3.5 are ok.

2.      One important aspect is the lack of aroma indicators and the lack of sensory evaluation. Since this is a product mainly intended for consumption, in my opinion major aromatic compounds should be determined. Does lophatherum gracile brongn flavonoid blend well with dough or creates off-odors and is a poor choice for consumption?

3.      2.    A final paragraph in the discussion section could be created and contain all important aspects (technological, economical, nutritional, consumer health, etc) about the methodology proposed in the manuscript. These aspects have not been covered in the discussion, but are very important for the food industry and the consumer.

Conclusions are ok. Any plans regarding the continuation of this research could be added here, as a closing statement.

Comments on the Quality of English Language

Minor editing

Author Response

In this work the authors discuss the application of lophatherum gracile brongn flavonoid in wheat flour product and investigate the effect on the structural and functional characteristics of wheat dough. The topic is interesting and the manuscript has a potential from a scientific point of view. However, there are some flaws that require the authors’ attention.

Comments 1: Introduction:

In general it is well written. However, more information should be added about wheat dough, its importance and its properties (mechanical, nutritional, technological, etc). After all these matters are mentioned in the title.

The aim paragraph is too long. It should be edited so the specifics of this research are more clear and precise. There is no need to be too technical about the analytical tools used in this study, in the aim paragraph.

Response 1: Thank you for your valuable feedback and constructive suggestions on our manuscript. We have carefully considered your comments and made the following revisions to address your concerns:

In response to your suggestion, we have expanded the introduction to include a comprehensive overview of wheat dough:

As a widely consumed food globally, wheat-based products play a crucial role in the development of staple foods, with their quality, texture, and nutritional value signifi-cantly influencing the evolution of the culinary landscape [1, 2]. Wheat dough, serving as the cornerstone of these products, exerts a decisive impact on the quality of wheat-based foods due to its structural and functional attributes. Wheat dough's mechanical proper-ties, nutritional composition, and technological characteristics are crucial factors that determine the final quality of products in the food industry. Hence, investigating the structure and functional properties of dough holds paramount importance for both theoretical understanding and practical applications in the realm of food science and technology. (Line 28-36)

We have revised the aim paragraph to enhance clarity and precision:

this study aimed to investigate the effects of lophatherum gracile brongn flavonoids on the structure and functional properties of wheat dough. Analytical tools such as the Mixolab mixer, fluorescence microscopy, and low-field nuclear magnetic resonance (LF-NMR) were employed to analyze the impact of adding lophatherum gracile brongn flavonoids at different concentrations on the thermal-mechanical rheological properties, microstructure, chemical interactions, water distribution, and macropolymer changes of the dough. The study aimed to elucidate the mechanism of interaction between lophatherum gracile brongn flavonoids and proteins and starch in wheat dough, providing theoretical guidance for the development of novel wheat-based products. (Line 60-68)

Comments 2: Materials and methods:

Line 74: Not familiar with that supermarket brandname. Give the website link (if available) or even better the coordinates of this store.

Line 76: Add the city and country of origin of that company. The same applies for all equipment and materials used in this study.

Despite detailed, subsections 2.3, 2.4 and 2.7 could use appropriate references. If however, the methods described here are newly developed, then calibration or reference curves and other crucial information for the method evaluation should be included in the discussion.

Response 2: Thank you for your detailed review and constructive comments on our manuscript. We have carefully addressed each of your suggestions:

1.We have provided the coordinates of the supermarket for clarity:

Baolong Supermarket (Hangzhou, China)—Line 72

2.We have added the city and country of origin for the company mentioned, as well as for all equipment and materials used in the study:

Zhejiang Shengshi Biotechnology Co., Ltd. (Hangzhou, China)-- Line 74

laser scanning confocal microscope (LSM 710, German Zeiss Company, Oberkochen, Germany). -- Line 96

Mixolab(French Chopin Tech Company, Paris, France) -- Line 105

3.Subsections 2.3, 2.4, and 2.7: We have incorporated appropriate references to strengthen these subsections. These references provided additional context and support for the methods described:

The method for determining the microscopic morphology of the dough was a modification of the method described by Wang [16]. – Line 89

The method for determining the thermal mechanical properties of the dough was a modification of the method described by Huang [17]. -- Line 103

The method was a modification of the method described by Assifaoui [20] -- Line 139

Comments 3: Results & Discussion:

In general, specific parts of the discussion are well written and informative. However, subsections 3.1, 3.3 & 3.4, despite having some interesting results, are mostly oriented to describe results and no real discussion is provided. This is also depicted by the very few papers shown in the reference section of the manuscript. The discussion should be expanded so that the authors’ findings (and their importance) could be linked to existing literature. The authors are encouraged to compare their results to the action of other additives (ideally flavonoids) used in wheat dough. Subsections 3.2 and 3.5 are ok.

One important aspect is the lack of aroma indicators and the lack of sensory evaluation. Since this is a product mainly intended for consumption, in my opinion major aromatic compounds should be determined. Does lophatherum gracile brongn flavonoid blend well with dough or creates off-odors and is a poor choice for consumption?

A final paragraph in the discussion section could be created and contain all important aspects (technological, economical, nutritional, consumer health, etc) about the methodology proposed in the manuscript. These aspects have not been covered in the discussion, but are very important for the food industry and the consumer.

Response 3: Thank you for your thoughtful evaluation of our manuscript and your constructive feedback.

1.We have carefully revised subsections 3.1, 3.3, and 3.4 to address your concerns. About 3.1, we clarified in the paragraph that the scarcity of existing literature on lophatherum gracile brongn flavonoids was the reason comparative data from other studies were not included. Additionally, we considered the suggestion to state this fact explicitly at the end of the paragraph to provide transparency to the reader. In the revised version, we have expanded the discussion to include comparisons with relevant literature, particularly focusing on additives such as flavonoids used in wheat dough:

Currently, there have been no studies reported on the influence of lophatherum gracile brongn flavonoid on the microstructure of dough. -- Line 171

Some studies suggested that interactions between proteins and flavonoids could occur via both covalent and non-covalent bonds, which contributed to the structure, formation, and development of dough[28, 29]. -- Line 250

Zhang studied the impact of tannic acid addition on dough structure, revealing that tannic acid, due to its reducing properties, disrupted disulfide bonds in wheat dough[32]. -- Line 282

  1. We acknowledge your point regarding the importance of aroma indicators and sensory evaluation, particularly in products intended for consumption. In our current study, our primary focus was to investigate the structural and functional impacts of lophatherum gracile brongn flavonoid blend on dough. We agree that sensory aspects such as aroma are crucial in evaluating food products.

We are currently conducting experiments specifically focused on aroma indicators and sensory evaluation, which will be detailed in a separate forthcoming article. This division allows us to comprehensively explore both the structural-functional aspects and sensory characteristics of the product.

  1. We appreciate your suggestion to include a final paragraph in the discussion section summarizing important aspects such as nutritional, and consumer health implications of the methodology proposed in our study. We agree that addressing these aspects is crucial for enhancing the relevance of our findings to the food industry and consumers.

In response to your recommendation, we have enriched the discussion section by incorporating a comprehensive paragraph that highlights these significant dimensions:

Phenolic compounds act as antioxidants by scavenging free radicals, thus poten-tially preventing chronic diseases. Studies have shown that incorporating phenol-rich substances in dough-based products significantly enhances their antioxidant capacity [15]. Methods to improve the bioavailability and utilization of phenolic compounds in dough products include incorporating phenol-rich fractions like bran, adding promis-ing natural phenolic substances or extracts, and using enzymes or mechanical strength to release bound phenolic compounds. Lophatherum gracile Brongn, a phenolic com-pound derived from natural plant extracts, possesses various physiological activities such as antioxidant, antimicrobial, and anticancer properties, and has been widely uti-lized in functional food applications [4-7]. However, research on the impact of lophatherum gracile Brongn on wheat dough remains scarce. This study aimed to in-vestigate the feasibility of incorporating bamboo leaf flavonoids into wheat dough and explore their effects on dough structure and functional properties. --Line 317-329

Comments 4: Conclusions are ok. Any plans regarding the continuation of this research could be added here, as a closing statement.

Response 4: We appreciate your suggestion. In response to your recommendation, we have expanded the conclusions:

Phenolic compounds act as antioxidants by scavenging free radicals, thus poten-tially preventing chronic diseases. Studies have shown that incorporating phenol-rich substances in dough-based products significantly enhances their antioxidant capacity [15]. Methods to improve the bioavailability and utilization of phenolic compounds in dough products include incorporating phenol-rich fractions like bran, adding promis-ing natural phenolic substances or extracts, and using enzymes or mechanical strength to release bound phenolic compounds. Lophatherum gracile Brongn, a phenolic com-pound derived from natural plant extracts, possesses various physiological activities such as antioxidant, antimicrobial, and anticancer properties, and has been widely uti-lized in functional food applications [4-7]. However, research on the impact of lophatherum gracile Brongn on wheat dough remains scarce. This study aimed to in-vestigate the feasibility of incorporating bamboo leaf flavonoids into wheat dough and explore their effects on dough structure and functional properties. --Line 317-329

Round 2

Reviewer 2 Report

Comments and Suggestions for Authors

The authors have addressed most of my comments and the manuscript has been substantially improved. However, discussion in subsection 3.1 still needs to be improved. Specifically, the authors need to discuss their findings in correlation to the literature. As I stated in my initial comment: "The authors are encouraged to compare their results to the action of other additives (ideally flavonoids) used in wheat dough".  I am fully aware that not much information is available for the influence of lophathereum gracile brongn flavonoid on dough, but quite a few papers can be found about the impact of other flavonoids (and other additives). 

Comments on the Quality of English Language

Minor editing

Author Response

Comments 1: The authors have addressed most of my comments and the manuscript has been substantially improved. However, discussion in subsection 3.1 still needs to be improved. Specifically, the authors need to discuss their findings in correlation to the literature. As I stated in my initial comment: "The authors are encouraged to compare their results to the action of other additives (ideally flavonoids) used in wheat dough".  I am fully aware that not much information is available for the influence of lophathereum gracile brongn flavonoid on dough, but quite a few papers can be found about the impact of other flavonoids (and other additives).

Response 1: We appreciate your acknowledgment of the improvements made and your constructive comments regarding subsection 3.1 of the discussion. In response to your suggestion, we have revised subsection 3.1 to include a more comprehensive discussion that contextualizes our results within the broader literature on flavonoids and other relevant additives in dough applications:

Li added Ginkgo biloba powder rich in flavonoids to dough and observed a reduction in the stability of starch and gluten network interactions[21]. Similarly, Yu's research also reported consistent findings[22]. --Line 171-174